# Psychological Wellbeing in Adolescents with Leukaemia: A Comparative Study with Typical Development Peers

**DOI:** 10.3390/ijerph17020567

**Published:** 2020-01-16

**Authors:** Marta Tremolada, Livia Taverna, Ilaria Tamara Chiavetta, Sabrina Bonichini, Maria Caterina Putti, Alessandra Biffi

**Affiliations:** 1Department of Development and Social Psychology, University of Padua, 35131 Padua, Italy; ilariatamara.chiavetta@studenti.unipd.it (I.T.C.); s.bonichini@unipd.it (S.B.); 2Department of Child and Woman Health, University of Padua, 35127 Padua, Italy; mariacaterina.putti@unipd.it (M.C.P.); alessandra.biffi@unipd.it (A.B.); 3Faculty of Education, Free University of Bolzano-Bozen, 39042 Brixen, Italy; livia.taverna@unibz.it

**Keywords:** children, adolescents, leukaemia, in treatment, healthy peers, life perceptions, hope, psychological wellbeing, cognitive problems

## Abstract

There is still little research on psychological wellbeing, life satisfaction and reported problems in preadolescents and adolescents under therapy for leukaemia, and also little research comparing them with their healthy peers. The present study aimed to analyse the life satisfaction, hope, psychological wellbeing and reported problems’ intensity in 60 patients aged 8–18 during the first year of therapy, to identify those more at risk and to compare their reports with matched healthy peers. A battery of self-reported questionnaires was administered during hospitalisation or day hospital admissions post 6 months and post 12 months from the diagnosis. Younger patients (aged 8–13 years) were more at risk than older ones in their problems’ intensity and psychological symptoms; females and Acute Myeloid Leukaemia patients reported lower current life satisfaction perceptions; hope was associated with lower depression symptoms and mood problems. Healthy peers have a better perception of current life, but reported a lower hope score, more anxiety symptoms and more cognitive problems than patients. The first 6 months were more critical for patients’ psychological health. Basing on these empirical data, the inclusion of mental health care professionals or supportive psychotherapy into the treatment is recognized as extremely useful.

## 1. Introduction

Adolescence is a period of significant physical and emotional changes, and a diagnosis of leukaemia during this time could have an important impact on psychological development of adolescents. There is little research on how preadolescents and adolescents feel about their lives in this illness experience—how it affects their current lives and how they plan their future. Previous studies have pointed out that they identify fewer goals, are more likely to have a health-related goal and less likely to have an intrapersonal or leisure purpose and rate their objectives as more achievable and supported, compared with healthy peers [1].

Adolescents with cancer represent a major challenge to healthcare professionals both because of their serious illness and because this event comes at a sensitive period of development, the controversial age of adolescence. Healthcare professionals should know which factors influence the preadolescents’ and adolescents’ psychological wellbeing, life satisfaction, hope and the intensity of reported problems to be able to support them in their everyday life, in facing the illness and in helping them in taking decisions regarding their own future.

Quality of life (QOL) during cancer treatment is an important aspect of patient care and healing, but research has mostly focused on childhood cancer survivors [2]. Few studies have taken into consideration the psychological adjustment in adolescents during treatment, reaching controversial findings. The more optimistic point of view, taken in some of the studies, revealed that the majority of these patients did not demonstrate significantly elevated levels of anxiety or depression, relative to age-matched controls [3] or longitudinally [4], with a minority of 17–30% demonstrating symptoms of depression and anxiety [5]. Abrams and colleagues [6] found no significant differences in the depression and anxiety ratings in the cancer group relative to age-matched, healthy controls, although a substantial number of patients in both groups showed elevated anxiety and depression scores. Only a minority of patients showed severe anxiety symptoms (13–28.6%) [7], and the majority of them tended to report average levels of positive affect and low levels of negative affect, compared to healthy populations [8].

Others studies, with a more pessimistic point of view, i.e., Wu et al. [9], comparing young individuals with cancer in therapy to those already in an off-therapy regimen and to healthy controls, found that the former reported poorer psychological functioning in the in-therapy phase. Most frequently reported cancer-specific psychosocial problems were worry (about relapse and side effects), cognitive problems (e.g., attention difficulty) and physical appearance (not looking attractive) [10]. The level of psychological symptoms was significantly higher in adolescents with leukaemia compared with healthy adolescents [11,12] with special attention to self-esteem and academic difficulties as cancer survivors [13]. Several studies found that adolescents with cancer had both physical and psychological symptoms, but that they had more psychological and school-related issues (such as interruption in their education, academic failure, not attending school activities), anxiety, somatisation, and depression [14] than their healthy peers.

Several studies identified risk factors for those adolescents who will have trouble in psychological wellbeing or life perceptions. Certain types of cancer seemed to be associated with higher psychological functioning, with a better achievement in adolescents who were diagnosed with haematological cancer [15]. Females also reported weaker psychological and cognitive functioning, with a poorer outlook on life, in childhood cancer survivors, when compared to healthy females [16]. At highest risk were female adolescents and, in general, late adolescents. Similar findings were reported in younger children aged 8–12 years [17].

A possible protective resource could be identified in hopefulness, predictive of a good health-related quality of life (HRQoL) in the near future [18]. This resource was correlated with a positive sense of wellbeing and commitment to treatment, and also improving coping and self-esteem, especially in females [19]. Relatively high rates of hopefulness were reported in the first six months of treatment. The adolescents who had positive expectations were able to focus on specific hopes as a way of adapting to their illness and were able to develop a better sense of being, which might lead to improved compliance. In addition, optimism was discovered as a key concept associated with higher HRQoL in adolescents with cancer [20]. They were not more optimistic than their healthy peers, but they were significantly less pessimistic. Optimism and pessimism were related to different aspects of wellbeing. Specifically, a cohesive pattern was found in which optimism predicted positive aspects and pessimism predicted negative aspects of wellbeing [21].

### Aims

The present study aims to analyse the life perceptions, hope, psychological wellbeing and reported problems in preadolescents and adolescents affected by leukaemia during the first year of therapy comparing perceptions and symptomatology with those reported by a group of matched healthy peers. There is no literature specific to the population under study, but focused principally on childhood cancer survivors.

The first area of investigation (A) was concerned exclusively with the patients’ group: we wanted to understand if the socio-demographic variables were associated with life satisfaction, hope, psycho-social wellbeing and cognitive functioning of the children and adolescents under therapy for leukaemia. Another aim was to understand the role of hope towards the symptomatology and if this symptomatology remained stable over time. We expected that:females were more likely to suffer emotional stress and to develop greater psychological symptomatology (A1);patients diagnosed with cancer in early adolescence declared a worse psychological adaptation than those who received the diagnosis during childhood [22], this difference will be maintained also in the older adolescents (A2);the economic condition would not be associated with life perceptions, psychological wellbeing and reported problems, following the analogous trend of other Italian studies on Adolescents and Young Adults (AYA) cancer survivors [23,24] (A3);the worst type of leukaemia (AML) was associated with a lower life satisfaction and more psychological problems, considering that type of illness could affect patients′ perception of wellbeing [25] (A4);there would possibly be positive associations of significance between hope and psychological functioning (A5), as the literature above had suggested to us;we would find better life perceptions, fewer reported problems and less symptomatology in patients at one year from the diagnosis than the same measures assessed at six months from diagnosis (A6).

The second area of investigation (B) concerned the comparison between patients and healthy peers with respect to their answers to the self-report questionnaires. Studies concerning the life perceptions and hope of patients compared to control groups are relatively few and controversial, so we aimed to understand whether and how patients in the first year of treatment have a different perception of life compared to healthy peers (B1).

The second question of area B (B2) focused on the symptomatology situation and the self-reported problems. Prior studies on this issue were conducted mainly on survivors and obtained contradictory results. Studies on a group of survivors reported symptoms of depression, somatic stress and self-esteem difficulties, especially if they underwent Haematopoietic Stem Cell Transplantation [23]. Moreover, another Italian study underlined better health-related quality of life, psychological wellbeing and cognitive functioning in AYA cancer survivors than in healthy peers [16], with also a positive sense of post-traumatic growth [24]. We also expected to find the same trend in our results, with more somatic complaints in the clinical group, but with lower reported cognitive problems and a high degree of hope.

## 2. Materials and Methods

### 2.1. Procedure

The present study was part of a major research project entitled: “Family factors predicting the short- and long-term adaptation and quality of life in children with leukaemia” approved by psychological research Ethical Committee, 2313 protocol number. All eligible adolescents attending the Paediatric Hematology-Oncologic Clinic at the University of Padua (Italy) in the period April 2017–April 2019 were asked to take part in this project. The written consent form was filled in by parents. In this study, patients answered a battery of self-reported questionnaires at two time points: the first (T1) was following the reinduction chemotherapy phase for Acute Lymphoblastic Leukaemia (ALL) (response therapy with risk band assignment) and the pretransplant phase for AML, approximately 6 months from diagnosis; the second time point (T2) occurred about the time of school re-entry of the child/adolescent, during the maintenance phase, one year after diagnosis. The battery of questionnaires analysed the perception of life, hope, psychological wellbeing and reported cognitive difficulties of children/early adolescents (8–13 years) and middle adolescents (14–18 years). We focused on these two age brackets of patients under treatment because from the developmental point of view, they have different aims to gain and consequently, they could have different behaviours in their life perceptions, psychological wellbeing and cognitive functioning. Patients that participated to this study were different between T1 and T2, so we showed data both separately, to have a picture of each time point, and also together to compare them with healthy peers.

Data were collected by a clinical psychologist and research assistant at the Haematology–Oncology Clinic, during patients′ hospitalisation or during out-patient admissions. In parallel, we obtained the self-report measures from the control sample of healthy peers. Control group participants (N = 60) met the following eligibility criteria: no history of life-threatening or chronic illness or injury and an absence of learning or sensory problems and other pathological aspects. They were enrolled in secondary schools, youth groups, and universities in the same region as the patients (Veneto, Northeast of Italy).

### 2.2. Participants

Patients (N = 60) were Caucasian with a mean age of 13.14 (SD = 2.86). Table 1 and Table 2 show the socio-demographic characteristics of the children/adolescents involved in the study and their parents, respectively, 6 months (N = 31) and 12 months (N = 29) after diagnosis. Furthermore, as a control group, healthy children/teens and their families were recruited and assessed with the same battery of instruments, to statistically compare the two populations.

Patients were matched by age, gender, family composition and perceived economic situation with the group of healthy peers, obtaining 60 pairs.

### 2.3. Instruments

#### 2.3.1. Brief Symptom Inventory 18

The Brief Symptom Inventory 18 (BSI-18 [26]) consists of 18 items grouped into three scales, serving as a screening for depression, somatisation and anxiety. Respondents are asked how they felt during the last 7 days, and each item is rated on a 5-point Likert scale from 0 (not at all) to 4 (extremely). BSI-18 was previously used to assess psychological outcomes in long-term survivors of childhood cancer [16] and in mothers of children under treatment for leukaemia [27,28]; demonstrating good internal consistency (Cronbach’s alpha ranged from 0.83 to 0.92). In this case, the adopted version is that for childhood cancer survivors.

#### 2.3.2. Socio-Demographic and Medical Questionnaire

This questionnaire is used for the collection of information useful for understanding the socio-economic and cultural level of the family. There are questions investigating parents′ age, the number of years of schooling and the qualification achieved, the work situation before and after the disease, demographic data of the family (family composition, number of children, housing and economic status, place of residence) and time availability for the child.

Medical information concerned: type of leukaemia, therapeutic protocol and effects of the therapies; number of days of hospitalisation; number of blasts at crucial moments of assessment; risk level (Standard Risk, Medium Risk, High Risk); degree of toxicity of the disease.

#### 2.3.3. Cognitive Problems Scale

This is a 25-item questionnaire that investigates the presence and intensity (range from 1 = ‘never a problem’ to 3 = ‘often a problem’) of cognitive problems shown by childhood cancer survivors [16] or reported by mothers of children under treatment for leukaemia in the last 2 weeks [27]. The CPS has been administered to 118 Italian parents of children with cancer, demonstrating global internal consistency *(alpha* = 0.89). A Varimax rotated confirmatory factor analysis, explaining a good proportion of the total variance (56.63%), identified five subscales: Memory (5 items; *alpha* = 0.78); Mental Disorganisation (8 items; *alpha* = 0.82); Labile Mood (3 items; *alpha* = 0.75); Impulsivity (4 items, *alpha* = 0.73); Concentration (5 items; *alpha* = 0.67). These five dimensions can be combined into a total score, the Cognitive Problems score (25 items; *alpha* = 0.89). The CPS is derived from the Childhood Cancer Survivor Study battery, and the wider purpose of this measure is to assess the frequency of possible cognitive problems that may arise in people that are under huge stress.

#### 2.3.4. Ladder of Life

The children/adolescents have to evaluate, using a 1–10 points scale, the quality of their present life and how satisfying their life will be in the future (5 years later), a sort of degree of hope for the future. With this instrument, we can obtain information about individual perception of the present and the future. It has been administered to 118 Italian mothers of children with cancer, demonstrating good global internal consistency (Cronbach’s alpha = 0.73) [28] and also to childhood cancer survivors [16].

### 2.4. Statistical Analysis Plan

To answer the developed research questions, several statistical analyses were carried out. Firstly, it was checked whether questionnaire scores differed within the clinical group with respect to gender (A1), age groups (A2), economic condition (A3) and type of leukaemia (A4), using a series of independent sample *t*-tests or ANOVAs. Subsequently, through Pearson′s correlations, we aimed at verifying if there were significant associations between the degree of hope and psychological functioning (A5). Independent-sample *t*-tests were run to understand the possible mean differences between symptomatology, life perceptions, hope and reported problems, between patients at T1 and those at T2 (A6).

To test aims B, sample-paired *t*-tests were performed to find out possible means differences in the scales of life perception (B1), the BSI-18′s symptomatology and cognitive problems (B2), between patients and healthy peers.

## 3. Results

### 3.1. The Impact of Socio-Demographic Variables on Patients’ Life Satisfaction, Hope, Psycho-Social Wellbeing and Cognitive Functioning

In the patient sample, we obtained a significant gender difference exclusively in the current life perception (t = 2.29, df = 57, *p* = 0.02), with females reporting lower scores (M = 6.48, SD = 2.22) than males (M = 7.66, SD = 1.43).

Regarding possible differences due to age, the results show mean differences of significance for Cognitive Problems total score (t = 3.04, df = 56, *p* = 0.004), disorganisation difficulties (t = 2.48, df = 56, *p* = 0.01), concentration (t = 2.6, df = 56, *p* = 0.01) and memory problems (t = 2.36, df = 56, *p* = 0.02), all of them higher for younger patients than for older ones, as shown in Figure 1. For the other scales regarding life perceptions and symptomatology, the results did not obtain a mean difference of significance (*p* > 0.05).

With respect to the perceived economic condition, the only significant difference was found for the sum of reported problems (F_2_ = 3.89, *p* = 0.02) that, adopting the Bonferroni correction, resulted in higher values in patients with a good economic condition than those with a medium one (mean difference = 0.37, *p* = 0.02).

Finally, we obtained a unique difference of significance between means of current life perception along patient’s type of leukaemia (t = 2.81, df = 57, *p* = 0.007), with patients affected by AML declaring a lower score (M = 5.71, SD = 2.27) than those with ALL (M = 7.35, SD = 1.78).

Then, hope was associated significantly with a lower depression symptomatology, assessed by BSI-18 (r = −0.3, *p* = 0.03) and lower reported mood problems, assessed by the Problem scale questionnaire (r = −0.3, *p* = 0.02).

### 3.2. Symptomatology, Life Perceptions, Reported Problems: Are there Mean Differences Comparing Patients’ Reports at T1 with Those at T2?

Mean differences of significance were obtained between the two patient groups for the following dependent variables: BSI-18 total symptomatology score (t = 2.96; df = 57; *p* = 0.004), BSI-18 depression symptoms (t = 3.04; df = 57; *p* = 0.004), BSI-18 somatisation symptoms (t = 2.55; df = 57; *p* = 0.01), reported mood problems (t = 2.43; df = 57; *p* = 0.01). Figure 2 shows these results.

### 3.3. Comparison of Scores Reported by Patients and Healthy Peers in the Different Scales in the First Year of Treatment

Current life perception and hope were different between the two groups (t = −2.45; df = 58; *p* = 0.01; t = 2.48; df = 58; *p* = 0.01). Specifically, current life perception was reportedly higher in the healthy group than in the patient group, while hope was higher in patients (Figure 3a).

Healthy peers reported BSI-18 anxiety higher scores comparing with patients’ scores (t = −3.9; df = 58; *p* = 0.0001) and likewise for total cognitive problems (t = −3.67; df = 58; *p* = 0.001). Specifically, problems reported as higher in the control group were impulsivity (t = −2.58; df = 58; *p* = 0.01), disorganisation (t = −3.12; df = 58; *p* = 0.003), concentration (t = −4; df = 58; *p* = 0.0001) and memory (t = −4.13; df = 58; *p* = 0.0001) (Figure 3b).

## 4. Discussion

Despite the increase in survival rates, cancer applies a physical, social and psychological cost both for the adolescent patients and for their families. One of the challenges health professionals are required to respond to is helping patients maintain physical health and psycho-social wellbeing [29]. This study aimed to fill some gaps in the literature with respect to the implications at the psycho-social level within the first year after the communication of the leukaemia diagnosis, thus in both the acute and chronic phases of therapies.

### 4.1. The First Gap Deals with Possible Socio-Demographic and Illness Differences in Patients’ Psychological Wellbeing such as Gender, Age at the Diagnosis, Perceived Economic Condition and Type of Leukemia

The international literature analyses the influence of socio-demographic variables on psychological wellbeing, stressing how patient females reported weaker psychological, social and cognitive functioning with a poorer outlook on life [15], also when compared to healthy females [10] and even in AYA cancer survivors [16]. However, no significant differences in any of the three symptomatology scales and cognitive problems emerged in the present study. The only difference was that females had a lower current perception of life than males, but their priorities and expectations could be different. It seems that prior to an event such as leukemia, boys and girls are more homogeneous in their reports.

Another aspect that affects the levels of perceived stress is the age at the time of diagnosis: those diagnosed with cancer in early adolescence seem to declare a worse psychological adaptation than those who received the diagnosis at an earlier age [22]. In this study, patients belonging to the age band 8–13 have shown higher anxiety, impulsivity and more intensive problems in mood, disorganisation and memory domains. This result could be due to the fact that older children might benefit from having more problem-solving strategies to cope with the illness and might have more communication with adults (parents, health professionals, other ill adolescents), compared with the preadolescents. This result is not in line with the literature that declared that later adolescents reported a worse condition and future studies with an ampler sample could better clarify this phenomenon.

Because the literature revealed that perceived economic condition is closely associated with higher levels of personal growth [30], this factor was investigated. We did not obtain relevant results, only that adolescents with good familial economic conditions reported more cognitive and mood problems than those belonging to a medium one. It could be that families might put more pressure on adolescents regarding these aspects, or it could be that patients, considering these capacities to be important, self-reported a lower ability.

Finally, the type of leukaemia emerged as an important factor, particularly for AML patients, who declared a lower life perception than ALL patients, probably because of the greater toxicity of this therapy, confirming also the major feeling of uncertainty in these patients. Hope was identified as a key factor related to fewer depression symptoms and mood problems, confirming its association with a positive sense of wellbeing and commitment to treatment, in the first year of therapies.

### 4.2. Psychological Wellbeing of Patients Compared with Healthy Peers along Timing

The second major focus of the present study was to compare the patients’ self-reports with those of matched healthy peers. The few studies that have assessed psychological health in active-treatment adolescents showed ambiguous results, stressing particularly more somatisation symptoms, but on the emotional side, there was not a unique response. In this study, throughout the first year post diagnosis, it emerges that patients perceived the present life worse than that of healthy peers, but they declared a higher degree of hope, fewer anxiety symptoms, less impulsivity and disorganisation, and fewer concentration and memory problems. These data can be explained if we consider the different daily life of these adolescents, made up of invasive therapeutic protocols, limitations, and isolation from peers, even if their emotive balance and cognitive functioning remained stable, attesting also at a best level compared with healthy peers, probably also due to their resilience skills.

This result could also be observed as a timing change: the emotive situation of preadolescents and adolescents ameliorated throughout time, showing, at 6 months post-diagnosis, more symptomatology and lower life perceptions for the present and for the future, while at 12 months post-diagnosis, there was a decrease in these indexes. This is a period in which they probably start to feel better, even physically, thanks to the maintenance therapy and they could return to their daily activities, such as school. They are more hopeful for the future and they reconsider their lives positively. Probably this positive trend, and the reappropriation of their normality, led patients to report fewer symptoms and fewer problems in their lives than healthy peers. Patients were also carefully followed by mental health professionals and by teachers, individually or as a group, to empower their coping strategies and cognitive functioning during the treatment cycles. The results obtained in the present research are in line with the optimistic literature (i.e., [16]), since children and adolescents in therapy declare fewer cognitive problems, so that they record lower scores than healthy peers in all the assessed reported problems, both emotive and cognitive.

### 4.3. Strengths and Limits of the Study

As this is a research project carried out at a single centre, it would be interesting to see what happens in other centres, in order to find further significant factors influencing patients’ psychological wellbeing. Moreover, it would also be interesting to widen assessments, for example, using not only self-reports, which could be influenced by social desirability. One of the disadvantages related to self-report measures is that they do not allow an objective assessment of the real cognitive performance of a person, because they are based on individuals’ perceptions of personal abilities. Social desirability bias could be partially avoided by introducing interviews with research participants. However, these problems are caused by ordinary limitations of clinical research.

Then, the SES data did not really add information to the final results. It feels as though the SES was assumed based on factors that may not be valid in the lived experience. A recommendation for future research could be to provide SES data that would consistently allow the application of social economic position of the family.

However, there are no known previous recorded Italian studies that have investigated these constructs in early and middle adolescents. Nevertheless, the present study was exploratory, and the sample involved had a too-limited number of participants. In fact, in some situations, the present study highlights results contrasting with other international research. The comparison with larger samples, based on multi-centre data collection, could lead to understanding if these results are attributable to cultural factors that deserve to be investigated.

The final aim of this project was to perform a specialised assessment of the various psychosocial aspects related to the wellbeing and life perception of these patients in order to identify those areas that are most problematic for them. Furthermore, developing interventions or remediation programs to target needs emerging in research studies like this one can lead to significant improvements as attested in other developmental domains [31]. Health personnel could be trained to prepare ad hoc interventions for preadolescents/adolescents such as teaching communicative strategies with younger adolescents and females to improve their hope or sustaining their parents in accompanying them during the acute treatment period. The primary focus of psychologists and psychotherapists should be on females, patients with AML, and preadolescents, especially during the first 6 months of therapies.

## 5. Conclusions

With regard to socio-demographic and illness variables, it is possible to conclude that they have a limited effect on patients’ psychological wellbeing, life perceptions and cognitive difficulties. Females and patients with AML were identified as at risk for their reported current life perception. On the other hand, age could influence the psychological wellbeing and reported intensity of problems of patients, with the 8–13 age group being more at risk. Comparing patients with healthy peers, this study showed that healthy peers have a better perception of current life, but lower hope, and a higher percentage of anxiety symptoms and cognitive problems than patients. The first 6 months could be critical for their emotional wellbeing, and hope could be considered a key element that provides an important influence on depression symptoms and mood states.

The inclusion of mental health professionals in the multidisciplinary treatment team could help in addressing potential difficulties in adjustment to the new life after the diagnosis. Early involvement in treatment may allow for earlier detection when problems arise and allow the patient and clinician to build the rapport and trust that is needed to address problems later on. Supportive psychotherapy, in the office or at the bedside, may be useful in addressing adjustment problems. The standard of health care in the Italian context could be unique and not similar in other international centres. However, we suggest to include the mental health professionals in any way in the treatment in the health care systems for leukemia patients.

## Figures and Tables

**Figure 1 ijerph-17-00567-f001:**
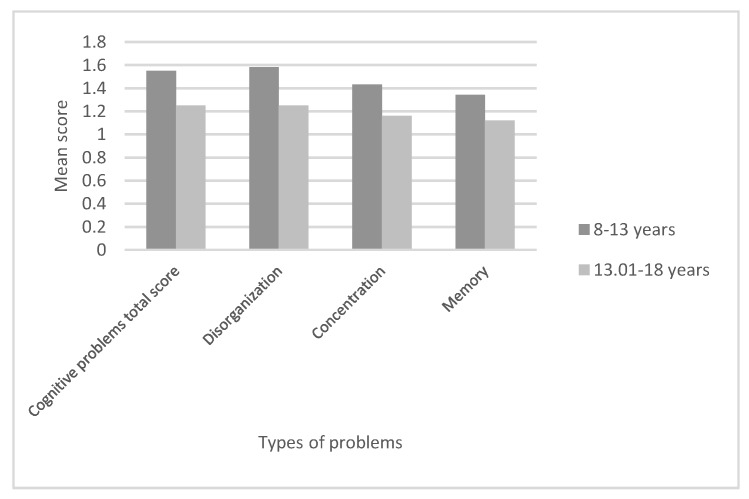
Cognitive problems (reported) according to patients’ age band.

**Figure 2 ijerph-17-00567-f002:**
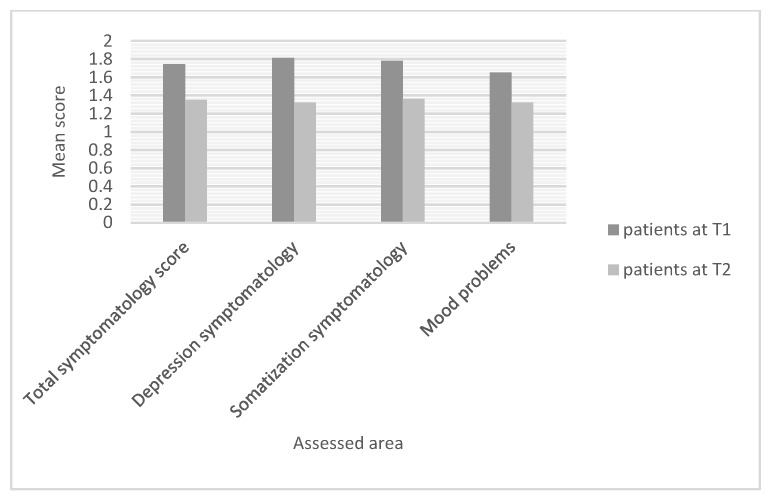
Patients’ symptomatology and reported problems at two time points.

**Figure 3 ijerph-17-00567-f003:**
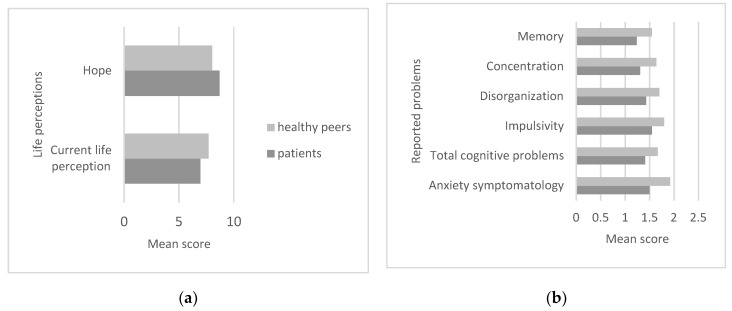
Mean differences of significance comparing patients and healthy peers regarding life perceptions (**a**), symptomatology and reported problems (**b**).

**Table 1 ijerph-17-00567-t001:** Socio-demographic characteristics of patients/families and healthy controls/families at 6 months (T1).

Child/Teen Socio-Demographic Characteristics		Control Group		Patients	
		Frequency	%	Frequency	%
Gender	Males	13	41.9	12	38.7
	Females	18	58.1	19	61.3
	Total	31	100	31	100
Age groups (years)	8–13	17	54.8	18	58.1
	13.01–18	14	45.2	13	41.9
	Total	31	100	31	100
		Mean	SD	Mean	SD
Current age		13.32	2.42	12.98	2.91
Type of leukaemia	Acute Limphoblastic Leukaemia			22	71.0
	Acute Myeloid Leukaemia			9	29.0
	Total			31	100
Family socio-demographic characteristics		Control group		Patients	
		Frequency	%	Frequency	%
Respondent parent’s educational level	Primary school	0	0	2	6.5
	Secondary school 1st grade	9	29	14	45.2
	Secondary school 2nd grade	9	29	11	35.5
	Degree	1	3.2	1	3.2
	Post Degree	7	22.6	3	9.7
	Missing	5	16.1	0	0
	Total	31	100	31	100
Respondent parent’s job	Job leave/housewife	1	3.2	14	45.2
	Abandonment/loss of work	4	12.9	5	16.1
	Part-time	7	22.6	10	32.3
	Full-Time	17	54.8	2	6.5
	Missing	2	6.5	0	0
	Total	31	100	31	100
Respondent parent’s workload	≥50 h/week	3	9.7	1	3.2
	40–49 h/week	5	16.1	8	25.8
	30–39 h/week	10	32.3	5	16.1
	20–29 h/week	5	16.1	6	19.4
	10–19 h/week	2	6.5	3	9.7
	0–9 h/week	2	6.5	8	25.8
	Missing	4	12.9	0	0
	Total	31	100	31	100

**Table 2 ijerph-17-00567-t002:** Socio-demographic characteristics of patients/families and healthy controls/families at 12 months (T2).

Child’s/Teen’s Socio-Demographic Characteristics			Control Group		Patients	
			Frequency	%	Frequency	%
Gender		Males	12	41.4	12	41.4
		Females	17	58.6	17	58.6
		Total	29	100	29	100
Age groups (years)		8–13	14	48.3	14	48.3
		13.01–18	15	51.7	15	51.7
		Total	29	100	29	100
			Mean	SD	Mean	SD
Current age			13.56	2.73	13.45	2.96
Type of leukaemia		Acute Limphoblastic Leukaemia			23	79.3
		Acute Myeloid Leukaemia			6	20.7
		Total			29	100
Family socio-demographic characteristics			Control group		Patients	
			Frequency	%	Frequency	%
Respondent parent’s educational level	Primary school		0	0	0	0
	Secondary school 1st grade		7	24.1	8	27.6
	Secondary school 2nd grade		12	41.4	16	55.2
	Degree		0	0	1	3.4
	Post Degree		7	24.1	4	13.8
	Missing		3	10.3	0	0
	Total		29	100	29	100
Respondent parent’s job	Job leave/housewife		1	3.4	15	51.7
	Abandonment/loss of work		2	6.9	4	13.8
	Part-time		6	20.7	7	24.1
	Full-Time		18	62.1	3	10.3
	Missing		2	6.9	0	0
	Total		29	100	29	100
Respondent parent’s workload	≥50 h/a week		3	10.3	1	6.9
	40–49 h/a week		7	24.1	6	20.7
	30–39 h/a week		10	34.5	4	13.8
	20–29 h/a week		3	10.3	3	10.3
	10–19 h/a week		1	3.4	5	17.2
	0–9 h/a week		1	3.4	9	31.0
	Missing		4	13.8	0	0
	Total		29	100	29	100

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
