# Peer review of "Psychological Wellbeing in Adolescents with Leukaemia: A Comparative Study with Typical Development Peers"

_ijerph, 2020, doi:10.3390/ijerph17020567_

Round 1

Reviewer 1 Report

This is a high quality paper and there are only minor suggestions for authors: 

Abstract:

Fulfil the number of participants inthe study into the abstract.

This part of the sentence: „After the parental written consent signature“ (21) seems to be not important for abstract. Omitting this you could spare more space for other information.

28-29: This sentence is not logical - it is in the final part of abstract that should report conclusions, not goals: „The clinical aim 29 was to identify the patients more at risk in order to prepare ad hoc psychological interventions.“ Furthermore, there is not clearly indicated in the article if this was done or not (was the intervention finally provided, what kind of intervention, etc.).

Also some clear conclusion is missing. I would suggest to replace the last sentence with a conclusion about the usefulness to include mental health care professionals or supportive psychotherapy into the treatment (as stated in conclusion section, 370-374).

Material and methods:

Could you please add some brief information about the Hospital/centre where data were collected in this section? This may be important since you mention this factor in the discussion (line 337). Eg. is it expected that the standard of health care will be similar in other centres? Are there mental health professionals included in any way in the treatment? – you mention this as a suggestion in the discussion. But is it a problem of your facility, the Italian health care system for leukemia patients or a problem internationally. There are various implications also for other sections, especially the introduction.

Discussion:

275-279: This part reminds more introduction that discussion. For the whole discussion section, it would be beneficial for readers, if you could use more sub-sections or highlight the important words (some map or graph concerning the important results would be excellent but not necessary).

351-359: This is not clear. You describe what you want to do, but not what was really done in this study. How this study relates to the specialised assessment? You suggest that health professionals could be trained (355-356) but are there some clearer suggestions from this study? In the next section you suggest the involvement of psychotherapists but this is not the same as to train the health personnel. This must be explained better.

The authors honestly reflect several other problematic aspects of the study (mainly limited demografic locality of data collection, sample size, etc.) that reduce the significance of its conclusions, but these problems are caused by ordinary limitations of clinical research. 

Moreover, I suggest to specify the identity of the grant in Funding section and to proof-read the article thoroughly (for instance: "Data were collected by a clinical psychology" - you probably meant "psychologist", line 146).

Author Response

First reviewer

This is a high quality paper and there are only minor suggestions for authors: 

Abstract:

Fulfil the number of participants in the study into the abstract.

This part of the sentence: „After the parental written consent signature“ (21) seems to be not important for abstract. Omitting this you could spare more space for other information.

28-29: This sentence is not logical - it is in the final part of abstract that should report conclusions, not goals: „The clinical aim 29 was to identify the patients more at risk in order to prepare ad hoc psychological interventions.“ Furthermore, there is not clearly indicated in the article if this was done or not (was the intervention finally provided, what kind of intervention, etc.).

Also some clear conclusion is missing. I would suggest to replace the last sentence with a conclusion about the usefulness to include mental health care professionals or supportive psychotherapy into the treatment (as stated in conclusion section, 370-374).

We revised the abstract along your suggestions. Thank you for your help. See lines 16-29.

Material and methods:

Could you please add some brief information about the Hospital/centre where data were collected in this section? This may be important since you mention this factor in the discussion (line 337). Eg. is it expected that the standard of health care will be similar in other centres? Are there mental health professionals included in any way in the treatment? – you mention this as a suggestion in the discussion. But is it a problem of your facility, the Italian health care system for leukemia patients or a problem internationally. There are various implications also for other sections, especially the introduction.

We added the requested information about the centre at lines 138-140. We also insert your comment in the conclusion section (See lines 409-411).

Discussion:

275-279: This part reminds more introduction that discussion. For the whole discussion section, it would be beneficial for readers, if you could use more sub-sections or highlight the important words (some map or graph concerning the important results would be excellent but not necessary).

We followed your suggestions adopting subheadings. See lines 284-383.

351-359: This is not clear. You describe what you want to do, but not what was really done in this study. How this study relates to the specialised assessment? You suggest that health professionals could be trained (355-356) but are there some clearer suggestions from this study? In the next section you suggest the involvement of psychotherapists but this is not the same as to train the health personnel. This must be explained better.

We clarified better this point at lines 378-383.

The authors honestly reflect several other problematic aspects of the study (mainly limited demografic locality of data collection, sample size, etc.) that reduce the significance of its conclusions, but these problems are caused by ordinary limitations of clinical research. 

We added this comment in the limits section. See linew 362-363.

Moreover, I suggest to specify the identity of the grant in Funding section and to proof-read the article thoroughly (for instance: "Data were collected by a clinical psychology" - you probably meant "psychologist", line 146).

The grant was expired and for this reason we didn’t insert it in the Funding section, even if we had nominated it. We revised the editing and we previously submitted the manuscript under the revision of an English proofreading service

Reviewer 2 Report

This is a good piece of research. Largely it feels ready for publication with some minor revisions:

Please let us know the dates of data collection. This helps us to know how current or dated the data is. The literature review in the introduction is good - it might need a clearer statement that there is no literature specific to the population under study.  It is unclear that the data collected on parents really does provide us with SES data that would consistently allow the application of social economic position of the family. The explanation of that link, why that data accomplishes what is suggested and how the conclusions are made. In the final analysis, this data does not really add to the results. It feels as though the SES has been assumed based on factors that may not be valid in the lived experience. There is some minor editing

Author Response

Second reviewer

This is a good piece of research. Largely it feels ready for publication with some minor revisions:

Please let us know the dates of data collection. This helps us to know how current or dated the data is.

We added this information in the procedure section. See lines 138-140.

The literature review in the introduction is good - it might need a clearer statement that there is no literature specific to the population under study. 

We added this information at lines 97-98

It is unclear that the data collected on parents really does provide us with SES data that would consistently allow the application of social economic position of the family. The explanation of that link, why that data accomplishes what is suggested and how the conclusions are made. In the final analysis, this data does not really add to the results. It feels as though the SES has been assumed based on factors that may not be valid in the lived experience.

We insert your precious comment in the discussion in the limits section at lines 364-367.

There is some minor editing 

We revised the editing and we previously submitted the manuscript under the revision of an English proofreading service.